# Robust and Scalable Autonomous Reinforcement Learning in Irreversible Environments

**Sang-Hyun Lee**
Department of Automotive Engineering
Ajou University
Gyeonggi-do, South Korea
sanghyunlee@ajou.ac.kr

## Abstract

Reinforcement learning (RL) typically assumes repetitive resets to provide an agent with diverse and unbiased experiences. These resets require significant human intervention and result in poor training efficiency in real-world settings. Autonomous RL (ARL) addresses this challenge by jointly training forward and reset policies. While recent ARL algorithms have shown promise in reducing human intervention, they assume narrow support over the distributions of initial or goal states and rely on task-specific knowledge to identify irreversible states. In this paper, we propose a robust and scalable ARL algorithm, called RSA, that enables an agent to handle diverse initial and goal states and to avoid irreversible states without task-specific knowledge. RSA generates a curriculum by identifying informative states based on the learning progress of an agent. We hypothesize that informative states are neither overly difficult nor trivially easy for the agent being trained. To detect and avoid irreversible states without task-specific knowledge, RSA encodes the behaviors exhibited in those states rather than the states themselves. Experimental results demonstrate that RSA outperforms existing ARL algorithms with fewer manual resets in both reversible and irreversible environments.

## 1 Introduction

Reinforcement learning (RL) has demonstrated remarkable achievements in the field of robotics [16, 14, 8, 18]. However, most of these achievements rely on repeated resets between episodes to provide an agent with multiple attempts and unbiased experiences. While such resets are easily performed in simulated settings, they require substantial human intervention and lead to poor training efficiency in the real world [28, 6, 12]. Autonomous RL (ARL), which simultaneously learns how to solve a task and how to reset an environment, has gained significant attention as a promising alternative to reducing human intervention. The key idea behind ARL is to generate a curriculum that determines when to abort episodes and where to return an agent. For example, the curriculum encourages an agent to explore space near initial or goal states in the early stages of training and gradually move to more distant space as training progresses.

Recent ARL works have shown that their curricula enable an agent to learn diverse tasks with fewer manual resets [9, 31, 24, 26, 20]. However, these works require narrow support over the initial or goal state distributions to generate their curricula. These restricted distributions can lead to less robust performance. Furthermore, their curricula either do not consider irreversible states or rely on task-specific knowledge to identify them. Irreversible states are those from which an agent cannot recover without external intervention, such as when a vehicle is damaged in an accident or a manipulator pushes an object outside its workspace. Most real-world tasks involve such irreversible states, and the task-specific knowledge required to identify them is rarely available in practice.

39th Conference on Neural Information Processing Systems (NeurIPS 2025).

In this paper, we introduce RSA, a robust and scalable ARL algorithm that enables an agent to handle diverse initial and goal states and to detect and avoid irreversible states without task-specific knowledge. RSA generates a curriculum by identifying informative initial and goal states based on the agent's learning progress. We hypothesize that informative states are those that are neither overly difficult nor trivially easy for the agent under training. To detect and avoid irreversible states, RSA encodes the behaviors exhibited in such states rather than the states themselves. This is based on our observation that, while irreversible states may vary across tasks, the behaviors tend to follow common patterns.

The main contribution of this work is a robust and scalable ARL algorithm called RSA. We evaluate RSA against baselines on diverse navigation and manipulation tasks. Experimental results demonstrate that RSA generates a curriculum by identifying informative states based on the agent's learning progress, achieving better performance with fewer manual resets than the baselines in both reversible and irreversible environments.

## 2    Related Work

RL agents trained in simulation often struggle to perform well in the real world due to the fidelity gap between simulated and real-world environments. A straightforward approach to avoid this gap is to train agents directly in the real world. Chebotar et al. [6] combine model-based and model-free updates to learn manipulation tasks on a PR2 robot. Kendall et al. [12] demonstrate the first RL agent capable of driving a real-world vehicle along country roads. However, these works rely on manual resets after every episode, which require substantial human intervention. While several previous works employ additional instrumentation or scripted reset behaviors to automate environment resets, their reset strategies are task-specific and limited to particular scenarios [15, 30, 18, 29, 22].

ARL, which aims to learn without manual resets, has been actively studied in recent years [9, 27, 31, 23, 25, 24, 26, 13, 20]. Eysenbach et al. [9] train forward and reset policies simultaneously and generate a curriculum by early aborting forward episodes based on the reset value function. While their curriculum encourages agents to explore space they have already learned, Patil et al. [20] generate a curriculum that guides an agent to explore space they have not yet sufficiently learned. Both works assume an unimodal initial state distribution with narrow support, which can cause sub-optimal and non-robust performance. To address this limitation, Zhu et al. [31] introduce a random perturbation controller that discovers novel initial states, gradually broadening the support of the training distribution. Sharma et al. [24] encourage the agent to return to states from expert demonstrations, based on the hypothesis that such demonstrations provide a desired distribution over initial states. Although these works focus on ensuring diverse initial states, their curricula do not account for multiple goal settings, which makes their algorithms less scalable. To address this limitation, our work generates a curriculum that provides an agent with diverse initial and goal states by identifying informative states based on the agent's learning progress.

While most previous works assume that environments are reversible, our work aims to reduce manual resets in both reversible and irreversible environments. Eysenbach et al. [9] and Xie et al. [26] are most closely related to ours, as they focus on reducing manual resets in irreversible environments. Eysenbach et al. [9] leverage the state-action value function, which is trained with reset reward functions, as a metric to determine whether an agent is in irreversible states. Xie et al. [26] propose a label-efficient binary search algorithm that detects irreversible states using a limited number of reversibility labels. Experimental results from both works demonstrate that their algorithms can identify irreversible states and prevent agents from entering them. However, in real-world scenarios, access to task-specific knowledge, such as reset reward functions or reversibility labels, is often unavailable. Our work enables an agent to identify and avoid irreversible states without task-specific knowledge. Table 1 summarizes the key features that distinguish our work from previous works.

## 3    Preliminaries

### 3.1    Goal-Conditioned Reinforcement Learning

We consider a goal-conditioned reinforcement learning (GCRL) task represented by the tuple $(\mathcal{S}, \mathcal{A}, \mathcal{G}, \mathcal{T}, r_g, \rho_0, \gamma, \rho_g, m)$. $\mathcal{S}$ is the set of states $s$, $\mathcal{A}$ is the set of actions $a$, $\mathcal{G}$ is the set of goal states $g$, and $\mathcal{T} : S \times A \times S \to [0, 1]$ is the state transition model. $r_g : \mathcal{S} \times \mathcal{A} \times \mathcal{S} \to \mathbb{R}$ is the

Table 1: Comparison of Autonomous Reinforcement Learning (ARL) algorithms.

| Property | LNT [9] | R3L [31] | MEDAL [24] | PAINT [26] | RSA (Ours) |
|---|---|---|---|---|---|
| Diverse Initial States | ✗ | ✓ | ✓ | ✓ | ✓ |
| Diverse Goal States | ✗ | ✗ | ✗ | ✗ | ✓ |
| No Extrinsic Reset Rewards | ✗ | ✓ | ✓ | ✓ | ✓ |
| No Demonstrations | ✓ | ✓ | ✗ | ✓ | ✓ |
| No Reversability Labels | ✓ | ✓ | ✓ | ✗ | ✓ |
| Irreversible Environments | ✓ | ✗ | ✗ | ✓ | ✓ |

goal-conditioned reward function, $\rho_0$ is the initial state distribution, and $\gamma$ is the discount factor. $\rho_g$ is the goal state distribution, and $m : \mathcal{S} \to \mathcal{G}$ is a mapping function from states to corresponding goal states. The goal space may either be a subset of the state space or the state space itself. In the latter case, the mapping function is the identity function. An agent's behaviors are determined by a goal-conditioned policy and value function. The goal-conditioned policy $\pi(a \mid s, g)$ maps the current state and the current goal to a probability distribution over actions and the goal-conditioned value function $Q^\pi(s, a, g)$ represents the expected return when an agent takes action $a$ in state $s$ and goal $g$ and follows the policy $\pi$. The goal of GCRL is to find the optimal policy that maximizes the expected return when the state transition model is unknown. Please refer to [11, 21, 1] for further details.

### 3.2 Successor Features

Assume the reward function is a linear combination of a feature vector $\phi(s, a, s') \in \mathbb{R}^d$ and a weight vector $\omega \in \mathbb{R}^d$, such that the reward is given by $r(s, a, s') = \phi(s, a, s')^\top \omega$. Following [2, 17, 10], the goal-conditioned reward function can be similarly decomposed as $r_g(s, a, s') = \phi(s, a, s')^\top \omega_g$, as the weight vector, also referred to as a task vector, encodes preferences over individual feature components. With this formulation, the goal-conditioned value function can then be written as follows:

$$Q^\pi(s, a, g) = \mathbb{E}_\pi[\sum_{i=t}^{\infty} \gamma^{i-t} \phi_{i+1}^T \omega_g \mid S_t = s, A_t = a]$$

$$= \mathbb{E}_\pi[\sum_{i=t}^{\infty} \gamma^{i-t} \phi_{i+1} \mid S_t = s, A_t = a]^\top \omega_g = \psi^\pi(s, a)^\top \omega_g.$$

Barreto et al. [2] call $\psi^\pi(s, a)$ the Successor Features (SFs), which describe the expected discounted sum of features $\phi(s, a, s')$ under a policy $\pi$. The SFs $\psi^\pi(s, a)$ represent the expected discounted sum of features encountered when following policy $\pi$ and can be regarded as a multi-dimensional value function, where a feature vector acts as a reward function. This implies that SFs can be trained with standard reinforcement learning algorithms.

## 4 Robust and Scalable Autonomous Reinforcement Learning

RSA is designed to enable an agent to achieve robust performance with fewer manual resets in both reversible and irreversible environments. Similar to previous ARL algorithms [9, 31, 24, 26], RSA defines a forward policy $\pi_f(a \mid s, g)$ and a reset policy $\pi_r(a \mid s, g)$, and alternates between them. The forward policy is trained to solve tasks and the reset policy is trained to reset the environment. Note that each policy takes the current goal state as an additional input.

RSA considers two distinctive and challenging settings. First, RSA allows initial and goal states to be located anywhere in the state space $\mathcal{S}$. This contrasts with previous works that assume unimodal and narrow supports for the initial or goal state distributions to generate their curricula. Second, RSA assumes neither reversible environments nor access to task-specific knowledge for identifying irreversible states. This presents difficulties for previous works that rely on such task-specific knowledge, including reversibility labels or reset reward functions, to identify irreversible states.

To handle these settings, RSA 1) generates a curriculum by identifying informative initial and goal states based on the learning progress of an agent, and 2) identifies irreversible states by encoding the behaviors exhibited in those states without task-specific knowledge. In the remainder of this section, we describe how RSA generates the curriculum and identifies irreversible states in detail.

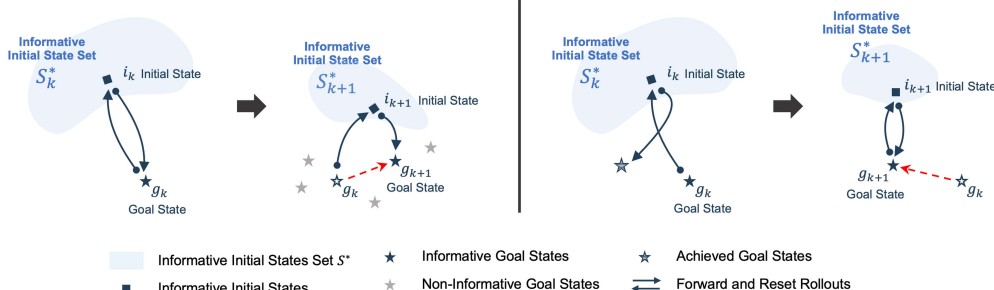

Figure 1: Curriculum generated by RSA. Given the current goal state $g_k$, the reset policy is activated until the agent reaches the informative initial state set $S_k^*$ corresponding to that goal. **Left:** At the end of each iteration, RSA samples the next goal state from the informative goal state set. **Right:** If the agent fails to reach the current goal state $g_k$, RSA uses the final state $s_T$ of the current iteration as the next goal state $g_{k+1}$.

## 4.1 Identifying Informative Initial and Goal States

Figure 1 illustrates how RSA generates a curriculum by identifying informative initial and goal states. The curriculum is built on our hypothesis that informative states are neither overly difficult nor trivially easy for the agent being trained. If the initial or goal states are too difficult, the agent may take unsafe actions and fail to reach the goal. On the other hand, the agent may struggle to obtain useful information even with sufficient time if the initial or goal states are too easy. Both cases can lead to suboptimal performance and poor sample efficiency. To identify informative states based on the agent's learning progress, RSA introduces the state information estimator, $I(s, g)$, which takes a pair of initial and goal states as input and estimates the probability of reaching the goal state from the initial state when the agent follows the forward policy being trained.

Here we describe how RSA utilizes the state information estimator to identify informative goal and initial states. RSA determines the informative goal state for the $k$ th iteration, $g_k$, as follows:

$$g_k \sim Unif(\mathcal{G}_k^*), \quad \text{where } \mathcal{G}_k^* \triangleq \{g \in \mathcal{G}_{1:k-1} \mid \lambda_1 \leq \mathbb{E}_{s \sim B_r}[I(s, g)] \leq \lambda_2\}. \quad (1)$$

$\mathcal{G}_k^*$ is the set of informative goal states for the $k$th iteration, $\mathcal{G}_{1:k-1}$ is the set of goal states obtained until the previous iterations, $B_r$ is the replay buffer for the reset policy, and $\lambda_1$ and $\lambda_2$ are the lower and upper thresholds of the reachability probability over the pairs of initial and goal states, respectively. This prevents an agent from being assigned goal states that are either too difficult or too easy, which allows it to obtain informative transitions. Specifically, a goal state is uniformly sampled from the set of informative goal states at the beginning of each iteration. If the agent reaches a goal state and it is still identified as part of the informative goal state set, RSA reuses it as the goal state for the next iteration rather than sampling a new one.

After the informative goal state $g_k$ is determined, RSA attempts to discover the informative initial state $i_k$ by activating the reset policy until the agent reaches the set of informative initial states $\mathcal{S}_k^* \subset S$ as follows:

$$\mathcal{S}_k^* \triangleq \{s \in S \mid \lambda_1 \leq I(s, g_k) \leq \lambda_2 \text{ for } g_k \sim Unif(\mathcal{G}_k^*)\}. \quad (2)$$

This prevents the agent from resetting with too challenging or too easy initial states. To continuously discover novel initial states, RSA trains the reset policy with an off-the-shelf exploration algorithm that guides an agent to under-explored states. While we implement the exploration algorithm using random network distillation (RND) [5] in our experiments due to its scalability and ease of implementation, RSA is compatible with any exploration algorithm [3, 19, 4].

Even when our curriculum provides an agent with informative initial and goal states, the agent being trained may still fail to reach the goal states. Repeated failures can cause the agent to be stuck in uninformative states. To address this challenge, when the agent fails to reach the goal state, RSA assigns the final state of the current iteration as the goal state for the next iteration. This idea is inspired by hindsight experience replay (HER) [1], which enables an agent to learn from undesired outcomes. The key distinction between RSA and HER is that, whereas HER treats the final state as the goal state within the same iteration, RSA uses the final state of the current iteration as the goal state for the next iteration.

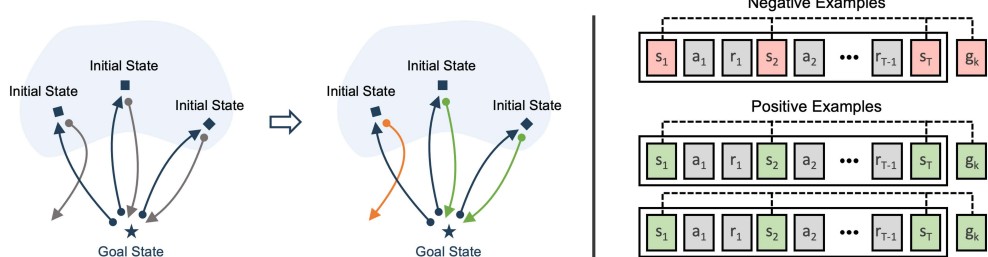

Figure 2: Overview of the training procedure for the state information estimator. State-goal pairs from successful forward rollouts are treated as positive examples, while those from failed forward rollouts are treated as negative examples.

The state information estimator, which estimates the probability of reaching goal states from initial states, is a key component that enables RSA to provide an agent with diverse initial and goal states. The main challenge in training the state information estimator is the absence of explicit supervisory signals to estimate reachability based on the agent's learning progress. To address this, RSA extracts reachability labels $e_t$ from forward rollouts and uses them to train the state information estimator in a self-supervised manner. Specifically, state-goal pairs from the rollouts in which the agent successfully reaches the goal state are treated as positive examples, while those from failed rollouts are treated as negative examples. The objective of the state information estimator can be written as follows:

$$\min_I -\mathbb{E}_{(s_t,g_t,e_t)\sim B_f}[e_t \log(I(s_t, g_t)) + (1 - e_t)\log(1 - I(s_t, g_t))], \qquad (3)$$

where $e_t$ is 1 for positive examples and 0 otherwise, and $B_f$ is the replay buffer for the forward policy. Figure 2 illustrates how RSA extracts supervisory signals from forward rollouts to train the state information estimator.

## 4.2 Detecting and Avoiding Irreversible States

RSA enables an agent to identify irreversible states without task-specific knowledge by encoding the behaviors exhibited in those states, rather than the states themselves. This is based on our observation that, while irreversible states have task-specific features, the behaviors tend to follow a shared pattern: an agent in irreversible states loses control over goal-relevant features, regardless of the actions it takes. To identify this shared pattern, RSA leverages SFs to encode behaviors induced by the reset policy. The SFs are denoted by $\psi^{\pi_r}(s, a) = \mathbb{E}\pi_r[\sum_{k=t}^{\infty} \gamma^{k-t}\phi_{k+1}|S_t = s, A_t = a]$. Note that the reset policy continues to generate diverse behaviors and the feature vector is defined as the norm of difference in goal-relevant features, $\phi_k = \|m(s_{k+1}) - m(s_k)\|$, where $m$ is the mapping function discussed in Section 3. RSA is agnostic to the form of the mapping function, and learning it with state-of-the-art representation learning is complementary to our work. However, we leave this for future work, as it is orthogonal to our main contribution.

We expect the SFs for irreversible state-action pairs to be much lower than those for reversible pairs. The set of irreversible states, $\mathcal{S}_{\text{irr}} \subset S$, can then be identified as follows:

$$S_{\text{irr}} \triangleq \{s \in S \mid \psi^{\pi_r}(s, a) \leq \lambda_3, \quad \text{for } a \sim \pi_r(a|s, g)\} \qquad (4)$$

where $\lambda_3$ is the reversibility threshold. RSA leverages two techniques to enable an agent to detect and avoid the set of irreversible states $S_{\text{irr}}$. First, RSA employs a surrogate reward function that penalizes the agent when it enters the identified irreversible states. This is similar to Eysenbach et al. [9] and Xie et al. [26], in which the agent also receives penalties for visiting irreversible states. However, unlike both previous works, RSA does not depend on task-specific knowledge to identify such states. Second, RSA conservatively identifies the irreversible states, by setting the threshold parameter $\lambda_3$ higher than that for the actual set, and aborts forward episodes when the agent enters any state in the identified set. We empirically found that this conservative identification helps prevent the agent from entering irreversible states in our experiments.

---

**Algorithm 1** Robust and Scalable Autonomous Reinforcement Learning

---

1: Initialize reset and forward policies $\pi_r(a|s,g), \pi_f(a|s,g)$
2: Initialize reset and forward replay buffers $B_r, B_f$
3: Initialize state information estimator $I(s,g)$ and successor features $\psi^{\pi_r}(s,a)$
4: **for** $k \leftarrow 1 \ldots K$ **do**
5:     Sample goal state $g_k$ with get_informative_goal$(B_r, \mathcal{G}_{1:k-1}, I(s,g))$
6:     **for** $t \leftarrow 1 \ldots T_{\text{reset}}$ **do**
7:         Select reset action $a_t \sim \pi_r(a_t|s_t, g_k)$
8:         **if** $\lambda_1 \leq I(s_t, g_k) \leq \lambda_2$ **then**
9:             Set $s_t$ as initial state and switch to forward policy
10:         **end if**
11:         Obtain reset transition $(s_t, a_t, r_t, s_{t+1}, g_k)$
12:         Compute surrogate reset reward $\hat{r}_t$
13:         Add transition to reset buffer $B_r \leftarrow B_r \cup \{(s_t, a_t, \hat{r}_t, s_{t+1}, g_k)\}$
14:         Update reset policy $\pi_r(a|s,g)$ and successor features $\psi^{\pi_r}(s,a)$
15:     **end for**
16:     **for** $t \leftarrow 1 \ldots T_{\text{forward}}$ **do**
17:         Select forward action $a_t \sim \pi_f(a_t|s_t, g_k)$
18:         **if** $\psi^{\pi_r}(s_t, a_t) \leq \lambda_3$ **then**
19:             Abort and switch to reset policy
20:         **end if**
21:         Obtain forward transition $(s_t, a_t, r_t, s_{t+1}, g_k)$
22:         Compute surrogate forward reward $\hat{r}_t$
23:         Add transition to forward buffer $B_f \leftarrow B_f \cup \{(s_t, a_t, \hat{r}_t, s_{t+1}, g_k)\}$
24:         Update forward policy $\pi_f(a|s)$ and state information estimator $I(s,g)$
25:     **end for**
26: **end for**

---

## 4.3 Algorithm Summary

Algorithm 1 outlines the overall training procedure of RSA. At the beginning of each iteration, RSA utilizes the state information estimator to identify a set of informative goal states from all goal states encountered in previous forward rollouts. It then samples one informative goal state from this set for the current episode. Once the goal state is determined, RSA activates the reset policy until the agent reaches an informative initial state, such that the predicted probability of reaching the goal state lies between $\lambda_1$ and $\lambda_2$. Upon reaching such a state, RSA designates the current state as the initial state and switches to the forward policy. When the forward policy is activated, RSA aborts the episode early if the SFs for the reset policy are less than or equal to the reachability threshold $\lambda_3$. Please refer to Appendix A for additional details.

## 5 Experiments

We design our experiments to investigate the following questions: (1) Can RSA achieve more robust performance with fewer manual resets than previous algorithms in both reversible and irreversible environments? (2) Can RSA generate a curriculum by identifying informative initial and goal states based on the learning progress of an agent? (3) How does each main component of RSA contribute to its performance improvement?

To answer these questions, we evaluate RSA against the following baselines: (1) LNT, (2) LNT-MG, (3) R3L, and (4) R3L-MG. LNT [9] uses reset reward functions to identify irreversible states and assumes narrow initial state distributions to generate its curriculum. R3L [31] generates a curriculum with diverse initial states but cannot handle irreversible states. Since neither algorithm considers multi-goal settings, we implement LNT-MG and R3L-MG as variants of LNT and R3L that sample goals randomly and periodically from their buffers. Note that RSA generates a curriculum that provides an agent with diverse initial and goal states and does not use reset reward functions to detect irreversible states. While all baselines perform manual resets either when the agent fails to return to starting states within a fixed number of episodes or enters irreversible states, RSA triggers manual

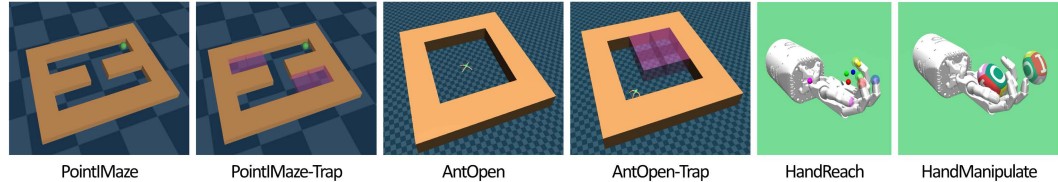

Figure 3: Navigation and manipulation tasks used in our experiments. PointIMaze and AntOpen are reversible, while PointIMaze-Trap and AntOpen-Trap include irreversible states, indicated by purple boxes. HandReach is reversible, whereas HandManipulate involves irreversible states, such as dropping the object from the hand.

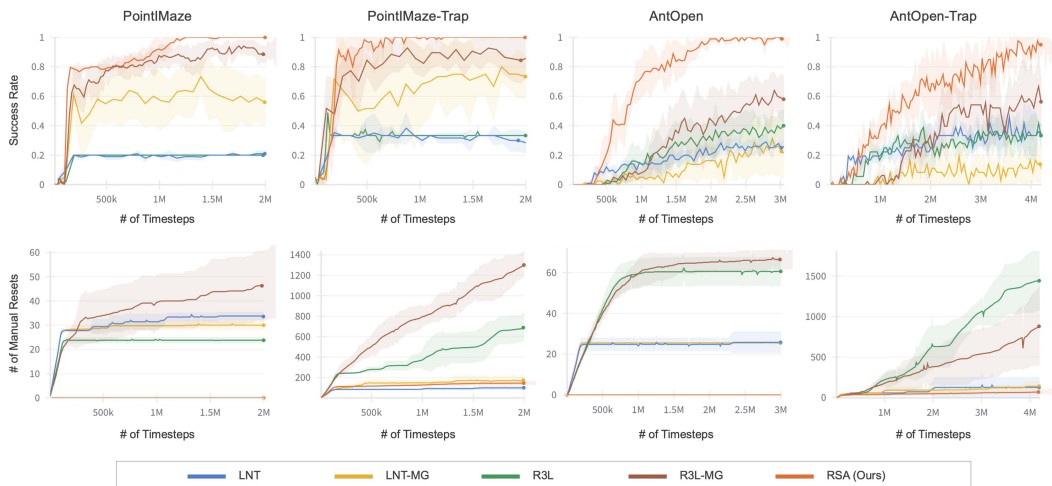

Figure 4: Learning curves for navigation tasks. The x-axis indicates the number of training steps, while the y-axis shows the success rate (top row) and the number of manual resets (bottom row). Darker-colored lines represent the means, and shaded regions denote the standard deviations across 5 random seeds.

resets only when the agent enters irreversible states. Appendix B provides additional details on our experimental setup and implementation.

## 5.1 Environments

Figure 3 shows the navigation and manipulation tasks used in our experiments. All tasks are provided by Gymnasium-Robotics [7]. The common objective across these tasks is to move an agent, such as a mobile robot, the fingertips of the hand, or an object, to target locations as quickly as possible. PointIMaze, AntOpen, and HandReach do not involve irreversible states, whereas PointIMaze-Trap, AntOpen-Trap, and HandManipulate contain such states. For example, when an agent in a navigation task enters a trap shown as a purple box, it struggles to move and cannot escape without external intervention. This is analogous to a vehicle damaged in an accident that is unable to move on its own. In the manipulation tasks, once an object is dropped, an agent can no longer pick it up or control it. For a fair comparison, we evaluate both the baselines and our algorithm using the same set of episodes with diverse initial and goal states. Appendix C provides additional details on the environments.

## 5.2 Experimental Results and Analysis

Figure 4 illustrates the learning curves computed over 5 random seeds for both reversible and irreversible navigation tasks. R3L and LNT achieve low success rates due to their narrow initial or goal state distributions. While R3L-MG and LNT-MG obtain robust performance in PointIMaze and PointIMaze-Trap, both suffer performance drops in AntOpen and AntOpen-Trap. We empirically observed that in AntOpen and AntOpen-Trap, agents tend to collect more biased experiences compared

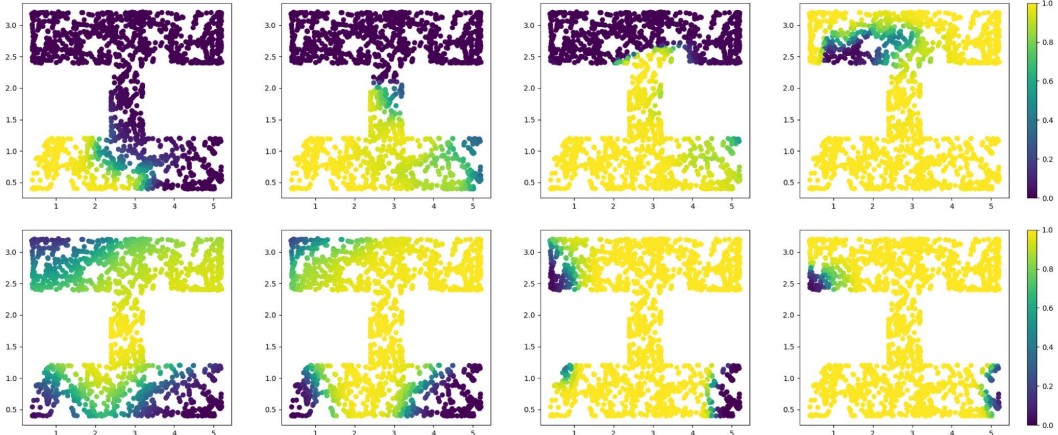

Figure 5: Changes in informative initial state sets over training. Each column shows how the informative initial states change as training progresses from left to right. The color of each initial state indicates its predicted reachability probability for a fixed goal state. **Top:** The goal state is located at $(0.8, 0.8)$. **Bottom:** The goal state is located at $(2.8, 1.8)$.

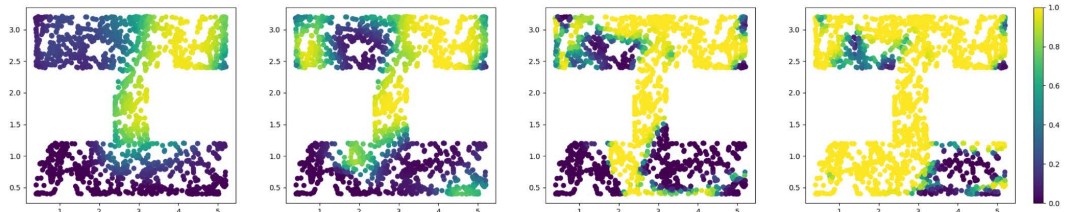

Figure 6: Changes in informative goal state sets over training. Each column shows how the informative goal states for a fixed initial state change as training progresses from left to right. The initial state is fixed at $(4.8, 2.8)$, and the color of each goal state indicates its predicted reachability probability from that initial state.

to PointIMaze and PointIMaze-Trap. This suggests that randomly sampling initial and goal states from the buffer fails to provide an agent with informative transitions when it struggles to explore diverse states. RSA achieves the best performance across all tasks and converges faster and more stably than the baselines.

LNT and LNT-MG require significantly fewer manual resets than R3L and R3L-MG. This result is expected, as LNT and LNT-MG assume narrow initial state distributions and leverage privileged information about irreversible states in the form of reset reward functions. However, constrained initial states make it difficult for an agent to collect diverse transitions, leading to suboptimal performance. Moreover, privileged information about irreversible states is rarely available in practice. RSA does not trigger manual resets in reversible navigation tasks, PointIMaze and AntOpen. This suggests that RSA can generate a curriculum that enables an agent to collect informative transitions even when it fails to return to initial states or reach to goal states. In the irreversible tasks, PointIMaze-Trap and AntOpen-Trap, RSA requires fewer manual resets than RSL and RSL-MG, and triggers a comparable number of resets to LNT and LNT-MG. These results indicate that, unlike LNT and LNT-MG, RSA can identify irreversible states without task-specific knowledge.

To investigate whether RSA identifies informative state sets based on the agent's learning progress, we visualize how the informative initial and goal state sets evolve during training on PointIMaze. Note that, in our work, informative initial and goal states are those whose estimated reachability probabilities fall between the hyperparameters $\lambda_1$ and $\lambda_2$. Figure 5 illustrates the changes in the informative initial state set for two different goal states: $(0.8, 0.8)$ and $(2.8, 1.8)$. In the early stages of training, the identified initial state sets are close to the respective goal states and gradually move farther away as training progresses. Figure 6 shows the changes in the informative goal state set for an initial state $(4.8, 2.8)$. We observed that similar to the informative initial state sets shown in Figure

5, the informative goal states are identified near the given initial state at the beginning of training and gradually spread out in diverse directions over time. These results demonstrate that RSA identifies informative initial and goal states based on the agent's learning progress. They also suggest that the curriculum generated by RSA enables an agent to efficiently bootstrap from success on easier initial and goal states to tackle more challenging ones. Appendix C provides additional results on the identification of irreversible states.

Finally, we conducted an ablation study to examine the benefits of two key components of RSA: (1) the state information estimator for identifying informative states, and (2) the successor features (SFs) for detecting and avoiding irreversible states. The left plot of Figure 7 shows the performance of RSA and its variant, RSA w/o SIE, which does not identify informative states, on HandReach. Note that in this reversible task, neither RSA nor RSA w/o SIE triggers manual resets, as both algorithms reuse the final state of the current iteration as the next goal state instead of resetting the environment. The performance gap between RSA and RSA w/o SIE indicates that identifying informative initial and goal states contributes to better asymp-

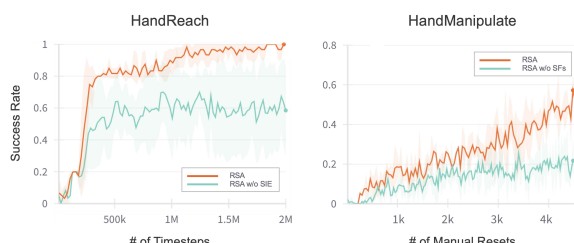

Figure 7: Ablation results of RSA components. The darker lines and shaded areas represent the mean and standard deviation over 5 random seeds. **Left:** Effect of identifying informative states on performance in HandReach. **Right:** Effect of identifying irreversible states on performance in HandManipulate.

totic performance and sample efficiency. The right plot of Figure 7 illustrates the performance of RSA and another variant, RSA w/o SFs, which does not identify irreversible states, on HandManipulate. Although both RSA and RSA w/o SFs can identify informative states using the state information estimator, we observed that RSA achieves better performance with the same number of manual resets. This result confirms that SFs of RSA play an important role in reducing manual resets.

## 6  Conclusion

We introduce a robust and scalable autonomous reinforcement learning (ARL) algorithm, referred to as RSA, that reduces manual resets in both reversible and irreversible environments. RSA identifies informative states to generate a curriculum that provides an agent with diverse initial and goal states. Furthermore, RSA enables an agent to detect and avoid irreversible states without task-specific knowledge. Experimental results demonstrate that RSA allows an agent to identify informative states based on the agent's learning progress and to avoid irreversible states, achieving better performance with fewer manual resets than existing ARL algorithms. RSA has several limitations that should be addressed in future work. First, RSA uses randomized reset behaviors, which are inefficient for discovering initial states in complex environments. We plan to investigate whether leveraging structured and consistent exploration behaviors can mitigate this limitation. Second, RSA uses predefined features to encode behaviors via successor features. We expect that combining RSA with state-of-the-art representation learning algorithms will be a promising approach to alleviating this limitation. Finally, RSA cannot prevent an agent from entering irreversible states before the associated behaviors have been encoded. We believe that the common-sense reasoning abilities of foundation models will make them promising for identifying irreversible states across diverse tasks, even in the early stages of training.

## Acknowledgements

This work was supported by Institute of Information & communications Technology Planning & Evaluation (IITP) under the Artificial Intelligence Convergence Innovation Human Resources Development (IITP-2025-RS-2023-00255968) grant funded by the Korea government(MSIT), Korea Ministry of Trade Industry and Energy, Korea Planning & Evaluation Institute of Industrial Technology(KEIT) under Grant RS-2024-00448797, and Convergence and Open Sharing System Project granted by the Ministry of Education and National Research Foundation of Korea.

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
