# OpenReview forum: "Robust and Scalable Autonomous Reinforcement Learning in Irreversible Environments"
_NeurIPS.cc/2025/Conference — NeurIPS 2025 poster_

### Official Review · Reviewer_tTbb · 2025-06-08

**Clarity:** 3
**Significance:** 3
**Originality:** 3
**Rating:** 4
**Confidence:** 3

**Summary:**

This paper addresses a key challenge in real-world reinforcement learning: the dependence on frequent manual resets of the environment after episode termination. This reset requirement is both labor-intensive and impractical in physical systems such as robotics.
The authors propose RSA (Robust and Scalable Autonomous RL), a method that trains both forward policies (to solve the task) and reset policies (to bring the environment back to favorable conditions) without task-specific knowledge and under diverse initial/goal states.
The paper is evaluated on continuous-control benchmarks including robotic manipulation and locomotion tasks, comparing RSA to state-of-the-art ARL methods such as SPOT and SkewFit.

**Questions:**

I think this is a **very thoughtful and practically valuable paper**. The motivation is crystal clear, and the solution is elegant and well-grounded. There’s a nice balance of algorithmic design and empirical demonstration.

That said, I would feel more confident in recommending a higher score if:

* There was a discussion of robustness to noise, and
* Some insight into the computational cost or training complexity was provided.

If these points are addressed, I would strongly lean toward an **accept** recommendation.

**Ethical Concerns:**

["NO or VERY MINOR ethics concerns only"]

**Limitations:**

Yes

**Quality:**

3

**Strengths And Weaknesses:**

**Strengths**:
I really like the idea of encoding how a state is reached instead of just looking at the state itself to judge irreversibility. This is quite clever—it acknowledges that the same final state can be reversible or not depending on the trajectory taken to get there, which is often true in manipulation tasks.

**Weaknesses**

**1. I’d like to see more theoretical insight**
While the empirical work is strong, I feel the paper would benefit from some **formal analysis**. For example:

* Is the learning progress metric provably helpful under certain assumptions?
* How does the behavior encoding affect reset success in theory?
  I’m not asking for complete proofs, but even a toy example or intuitive analysis would help deepen the reader’s understanding of *why* RSA works.

**2. I’m unsure how well this generalizes under noise**
One concern I have is how RSA performs in **stochastic or noisy environments**. If behavior encodings vary a lot between rollouts due to randomness, would the system still detect irreversible states reliably? I didn’t see this discussed in the paper. Including an experiment or a brief analysis on this point would strengthen the case for generality.

**3. Training complexity feels high but isn’t discussed**
RSA has multiple components (two policies, behavior encoders, progress estimators), and I wonder how **computationally heavy** the training is. This is especially relevant if someone wants to apply RSA to a real robot system. I would have appreciated a brief discussion of training time, or resource requirements.

---

> ### Author Rebuttal · Authors · 2025-07-31
>
> Thank you for your insightful and constructive feedback. We are pleased that you found the paper to be very thoughtful and practically valuable, the motivation to be crystal clear, the balance of algorithmic design and empirical demonstration to be nice, and the solution to be elegant and well-grounded. We would like to address your comments and questions below:
>
> ---
>
> **[W1] Intuitive analysis of why RSA works:** We definitely agree with your comment that an intuitive analysis would help deepen the reader’s understanding of why RSA works. RSA generates a curriculum by identifying informative initial and goal states based on the agent’s learning progress. We hypothesize that informative states are neither overly difficult nor trivially easy for the agent, which is motivated by the fact that **humans learn more effectively when tasks are provided in a meaningful order rather than at random, particularly when those tasks are around the edge of the current knowledge.**
>
> To detect and avoid irreversible states without task-specific knowledge, RSA encodes the behaviors exhibited in irreversible states rather than the states themselves. **In Appendix A, we provide a theoretical analysis demonstrating that the goal-conditioned value function for reversible states is greater than that for irreversible states when RSA penalizes visiting irreversible states.** We believe this theoretical analysis helps the readers understand how RSA encourages the agent to avoid irreversible states identified with the successor features.
>
> ---
>
> **[W2 & Q1] Robustness of detecting irreversible states to noise or stochasticity:** Thank you for your constructive feedback. We believe that RSA can effectively detect behaviors in irreversible states by using successor features even in stochastic environments. This is because the successor features in our work are trained to minimize temporal difference (TD) error on subsequent states. In other words, the successor features naturally take into account stochasticity in transitions by updating with expectations over transitions rather than deterministic transitions.
>
> Following your suggestion, **we implemented a new stochastic task called PointIMaze-Trap-Stochastic, a variant of PointIMaze-Trap, and conducted additional experiments.** In this task, the agent executes a randomly chosen alternative action with 25% probability instead of the action it originally selects. To provide a thorough analysis, **we visualized the irreversible states identified on both PointIMaze-Trap and PointIMaze-Trap-Stochastic in a manner similar to Figures 5 and 6 of the main paper.** We observed that, in PointIMaze-Trap, only states very close to the edges of the purple boxes are identified as irreversible. In contrast, in PointIMaze-Trap-Stochastic, states farther from the edges are also classified as irreversible. Given that uncertainty in the transitions expands the region where states are more susceptible to entering irreversible states, these experimental results suggest that RSA can robustly identify irreversible states even in stochastic environments. We will add the visualization results and the corresponding discussion to Appendix C in the revised paper. We hope this discussion addresses your concern.
>
> ---
>
> **[W3 & Q2] Resource requirements and computational costs:** We are glad to discuss the resource requirements and computational cost of RSA. Our experiments were conducted on two machines: a PC equipped with an RTX 5080 GPU and a workstation with four RTX 4090 GPUs. Each RSA run uses a fraction of a single GPU.
>
> We have to admit that RSA may require more computational cost than standard RL algorithms. This is because RSA trains two policies and two value functions to learn both how to solve tasks and how to reset environments simultaneously. In addition, RSA utilizes the state information estimator (SIE) to identify informative initial and goal states and the successor features (SFs) to detect irreversible states. Since the SIE and SFs have network structures similar to the policy and value function, respectively, RSA can be roughly considered as consisting of three policies and three value functions.
>
> **However, we would like to emphasize that the main bottleneck for training real-world robots is not that the computational cost is too high, but rather that manual resets entirely stop collecting data.** Our experimental results demonstrate that RSA requires significantly fewer manual resets even without task-specific knowledge. This indicates that RSA achieves better sample efficiency than previous state-of-the-art ARL algorithms in the real world. We will add this discussion to Appendix B in the revised paper. We hope this discussion addresses your concerns about resource requirements and computational costs.
>
> ---
>
> Thank you again for your time and effort in reviewing our work. We have carefully considered your valuable comments and tried our best to address your concerns with the clarifications and the additional new experiments. If our response has addressed your primary concerns, we would greatly appreciate it if you could update your score to reflect that. We are also willing to discuss with you if you have any further concerns.

---

> > ### Comment · Reviewer_tTbb · 2025-08-02
> >
> > Thanks for the response. At this stage, I will keep my score as is.

---

> > > ### Comment · Area_Chair_DT96 · 2025-08-05
> > >
> > > Dear Reviewer,
> > >
> > > Please kindly update your comment with more information about what (hypothetically) the authors could do to increase your score.
> > >
> > > Kind regards,
> > >
> > > --AC

---

### Official Review · Reviewer_wsCV · 2025-06-16

**Clarity:** 3
**Significance:** 3
**Originality:** 2
**Rating:** 4
**Confidence:** 3

**Summary:**

The manuscript proposes a robust and scalable autonomous reinforcement learning (ARL) algorithm, RSA, which addresses the limitations of existing ARL algorithms in handling diverse initial and goal states and avoiding irreversible states without relying on task-specific knowledge. RSA generates a curriculum by identifying informative states based on the agent’s learning progress and encodes behaviors to detect and avoid irreversible states. Experimental results show that RSA outperforms existing ARL algorithms in both reversible and irreversible environments with fewer manual resets.

**Questions:**

1. Why were other methods listed in Table 1, such as MEDAL and PAINT, not included in the experimental comparison? Are there technical or practical reasons for their exclusion?

2. In Section 5.2, why are experimental results on handReach and HandManipulate environments not reported for the comparsion?

3. In line 184, the authors expect that the successor feature values of irreversible state-action pairs to be much lower than those for reversible pairs. Is there theoretical justification or empirical evidence supporting this claim?

4. The explanations for experimental results are sometimes insufficient. For example, in line 227, why do narrow initial or goal state distributions lead to low success rates for R3L and LNT? In line 230, how is it observed that agents collect more biased experiences in AntOpen and AntOpen-Trap?

5. In line 243, it is mentioned that RSA and LNT require a comparable number of manual resets in irreversible environments. How does this demonstrate the advantage of RSA? Is LNT’s performance due to its use of task-specific knowledge, and how does this affect the fairness of the comparison?

**Ethical Concerns:**

["NO or VERY MINOR ethics concerns only"]

**Final Justification:**

Throughout the review process, my initial concerns centered on the lack of comprehensive empirical comparisons with all relevant baselines and the absence of quantitative analysis supporting a key design choice. The authors have responded to each point, conducting new experiments that directly address these issues. The expanded empirical evaluation now situates RSA more clearly within the broader ARL literature, and the quantitative analysis of successor features substantiates the method’s rationale. These improvements have shifted my opinion from a position of skepticism to one of support.

**Limitations:**

Yes

**Paper Formatting Concerns:**

There is a lack of detailed explanation or citation for the sub-algorithm in Algorithm 1 line 5, which may impede understanding.

**Quality:**

2

**Strengths And Weaknesses:**

### Strengths

1. The manuscript addresses a significant and practical problem in reinforcement learning, focusing on reducing manual resets and improving applicability in real-world settings.

2. RSA relaxes assumptions on initial and goal state distributions and does not require task-specific knowledge to avoid irreversible states, enhancing its generalizability and practical value.

3. The related work section provides a comprehensive comparison of ARL algorithms, and Table 1 clearly illustrates the advantages and coverage of the proposed approach.

4. The experimental results demonstrate RSA’s effectiveness and robustness in both reversible and irreversible environments, indicating its potential for real-world deployment.

5. The authors have provided the code, and the code is well-documented, facilitating reproducibility and independent verification of the experimental results.


### Weaknesses

1. The experimental comparison is limited, as it only includes two ARL baselines (and their multi-goal variants), omitting other relevant algorithms such as MEDAL and PAINT listed in Table 1. This limits the strength of the empirical validation.

2. The technical novelty of RSA appears incremental, and the method depends on three manually set threshold hyperparameters. The manuscript does not provide an analysis of the sensitivity or robustness of these parameters.

3. The experimental environments are relatively simple and mainly focus on navigation tasks, which may not sufficiently demonstrate the scalability and versatility of RSA in more complex or diverse domains.

---

> ### Author Rebuttal · Authors · 2025-07-31
>
> Thank you for your insightful and constructive feedback. We are pleased that you found the solved problem to be significant and practical, the algorithm to improve applicability and generalizability in real-world settings, the related work section to provide a comprehensive comparison of ARL algorithms, Table 1 to clearly illustrates the advantage and coverage of RSA, the experimental results to demonstrate the effectiveness and robustness of RSA in both reversible and irreversible environments, and the submitted code to be well-documented. We would like to address your comments and questions below:
>
> ---
>
> **[W1 & Q1] Comparison with other recent ARL algorithms (MEDAL and PAINT):** One of the most challenging aspects of designing our experiments was selecting baselines that can be compared fairly. In particular, unlike RSA, previous ARL algorithms require task-specific knowledge to reduce manual resets in reversible or irreversible environments. For example, MEDAL relies on expert demonstrations to generate its curriculum, and PAINT needs reversibility labels for all states to identify irreversible states. The performance of these algorithms can vary significantly depending on how such task-specific knowledge is constructed. For this reason, we selected ARL algorithms such as LNT and R3L, which leverage task-specific knowledge inherently available from the evaluation tasks, as baselines, while excluding algorithms like MEDAL and PAINT that require us to manually construct the task-specific knowledge they depend on.
>
> ---
>
> **[W2-1] Technical novelty of RSA:** We are glad to discuss the technical novelty of RSA. **To the best of our knowledge, RSA is the first autonomous RL algorithm that can reduce manual resets in both reversible and irreversible environments without task-specific knowledge.** The key ideas behind RSA are 1) generating a curriculum by identifying informative initial and goal states based on the agent’s learning progress, and 2) detecting and avoiding irreversible states by encoding the behaviors exhibited in those states rather than the states themselves. **These key ideas are fundamentally different from those of previous algorithms and are not merely incremental extensions.**
>
> ---
>
> **[W2-2] Analysis of the effects of hyperparameters introduced in RSA:** Thank you for your constructive feedback. To address your concern, **we conducted additional experiments on PointIMaze-Trap to examine the effects of our key hyperparameters.** First, we analyzed the performance according to the hyperparameters, $\lambda_1$ and $\lambda_2$, which are the lower and upper thresholds of the reachability range of informative initial or goal states. We observed that too difficult initial and goal states ($\lambda_1=0.0$ and $\lambda_2=0.1$) cause even more manual resets than random initial states. Conversely, too easy initial and goal states ($\lambda_1=0.9$ and $\lambda_2=1.0$) result in lower sample efficiency than the informative initial and goal states ($\lambda_1=0.1$ and $\lambda_2=0.6$). These results confirm the benefits of identifying informative initial and goal states.
>
> Next, we investigated the performance according to the reversibility threshold, $\lambda_3$. We observed that RSA maintains good performance and properly identifies irreversible states, except when the reversibility threshold is much higher than the output of the initialized successor features, $\psi^{\pi_r}(s,a)$, which causes the agent to classify all states as irreversible at the beginning of training. This discussion and additional experiments will be added to Appendix C in the revised paper.
>
> ---
>
> **[W3] Relatively simple experimental tasks:** Our experiments involve both reversible and irreversible tasks, consisting of four navigation tasks and two manipulation tasks. We would like to emphasize that, unlike standard reinforcement learning benchmarks, these tasks do not rely on periodic resets or provide access to any task-specific knowledge, such as reversibility labels or demonstration data. **These constraints make our tasks significantly more challenging than typical benchmarks, which may appear complex but are less demanding in practice.** In particular, one of our irreversible tasks, HandManipulate, is a well-known challenging task where standard RL algorithms struggle to learn even with periodic resets. We therefore believe that our experimental tasks are sufficient to demonstrate the scalability and versatility of RSA.
>
> ---
>
> **[Q2] Comparison results on HandReach and HandManipulate:** The reason we reported ablation studies on the key components of RSA instead of comparison results on HandReach and HandManipulate was to minimize redundancy and present as many diverse findings as possible within the page limit of the main paper. To be specific, the comparison results for these two manipulation tasks were largely consistent with those from the manipulation tasks, with the exception that all the baselines obtained extremely poor performance on HandManipulate. We will add the comparison results on both manipulation tasks to Appendix C in the revised paper.
>
> ---
>
> **[Q3] Empirical evidence supporting the claim that the SFs of irreversible pairs are much lower than those for reversible pairs:** Irreversible states are those from which an agent cannot recover without manual resets. As you mentioned, we claim that the successor features (SFs) for irreversible state-action pairs are much lower than those for reversible pairs. **This claim is based on our empirical observation that, in irreversible states, an agent loses control over goal-relevant features, regardless of the actions it takes.** For example, a vehicle damaged in an accident is unable to move toward its goal state by itself, and a manipulator cannot control an object once it leaves its workspace. Our experimental results on diverse irreversible tasks provide empirical evidence supporting this claim.
>
> Inspired by your constructive question, **we implemented additional evaluation code to visualize irreversible states in a manner similar to Figures 5 and 6.** The visualization results on PointIMaze-Trap and AntOpen-Trap show that at the beginning of training, all states are identified as reversible, as the state information estimator is initialized to output values higher than the reversibility threshold. As training progresses, irreversible states are gradually detected around the edge of the purple boxes, which represent the boundaries between reversible and irreversible regions of the state space. These results indicate that irreversible states identified by RSA align well with our intuitive notion of irreversibility. We will add the visualization results and the corresponding discussion to Appendix C in the revised paper.
>
> ---
>
> **[Q4-1] Low success rates due to narrow initial or goal state distributions:** The experimental results shown in Figure 4 describe that LNT and R3L achieve low success rates on reversible and irreversible maze navigation tasks. Several previous works support the interpretation that these poor performance is due to their narrow initial or goal state distributions. Sharma et al. [1, 2] empirically demonstrate that narrow initial state distributions lead to poor asymptotic performance and sample efficiency, as they constrain the state distributions visited by the agent. Kakade and Langford [3] provide a theoretical result showing that a more uniform initial state distribution can accelerate policy improvement by encouraging policy improvement at states that are unlikely to be visited. We will add this discussion to the first paragraph of Section 5.2 in the revised paper.
>
> ---
>
> **[Q4-2] More biased experiences in AntOpen and AntOpen-Trap:** LNT-MG and R3L-MG randomly sample initial and goal states, as these baselines cannot identify which states are informative. We empirically observed that in AntOpen and AntOpen-Trap, such random sampling leads to more biased experiences compared to PointIMaze and PointIMaze-Trap. Specifically, in PointIMaze and PointIMaze-Trap, the agent can collect diverse experiences even with randomly sampled initial and goal states due to its simple dynamics. In contrast, the agent in AntOpen and AntOpen-Trap struggles to collect diverse experiences unless informative initial and goal states are continously provided, as it has more complex dynamics. We will add this discussion to the first paragraph of Section 5.2 in the revised paper.
>
> ---
>
> **[Q5] Advantage of RSA over LNT:** Figure 4 shows that RSA and LNT require a comparable number of manual resets in irreversible tasks. However, we would like to note that RSA achieves better asymptotic success rates and sample efficiency than LNT. Furthermore, while LNT relies on task-specific knowledge to identify irreversible states, RSA identifies them without any task-specific knowledge. **These results suggest that RSA outperforms LNT under more challenging and general conditions.**
>
> ---
>
> Thank you again for your time and effort in reviewing our work. We have carefully considered your valuable comments and tried our best to address your concerns with the clarifications and the additional new experiments. If our response has addressed your primary concerns, we would greatly appreciate it if you could update your score to reflect that. We are also willing to discuss with you if you have any further concerns.
>
> [1] Archit Sharma, Rehaan Ahmad, Chelsea Finn. “A State-Distribution Matching Approach to Non-Episodic Reinforcement Learning.” International Conference on Machine Learning. 2012.
>
> [2] Archit Sharma, Kelvin Xu, Nikhil Sardana, Abhishek Gupta, Karol Hausman, Sergey Levine, Chelsea Finn. “Autonomous Reinforcement Learning: Formalism and Benchmarking.” International Conference on Learning Representations. 2022.
>
> [3] Sham Kakade, John Langford. “Approximately Optimal Approximate Reinforcement Learning.” International Conference on Machine Learning. 2002.

---

> > ### Comment · Reviewer_wsCV · 2025-08-01
> >
> > Thank you for your thorough responses to the previous concerns, as well as for conducting additional experiments and providing clarifications on the technical novelty and empirical robustness of RSA. Your detailed explanations and new results have addressed many of my initial questions, and I now have a deeper appreciation of the contributions. Nonetheless, I believe several important issues remain that warrant further consideration to strengthen the manuscript and its empirical claims:
> >
> > First, while I understand the practical difficulties in ensuring a fair comparison with algorithms such as MEDAL and PAINT due to their reliance on task-specific knowledge, I would like to reiterate that a comprehensive evaluation, including these methods under clearly stated assumptions, remains valuable. As you acknowledge, LNT and R3L also incorporate varying degrees of task-specific information, making absolute fairness elusive in any case. Reporting results for MEDAL and PAINT, even with explicitly documented procedures for constructing the required knowledge (e.g., using standard or minimally engineered demonstrations/labels), would provide a more complete empirical landscape. Such comparisons would not only highlight RSA’s unique strengths, particularly its independence from task-specific priors, but also clarify the practical trade-offs practitioners may face when some form of task knowledge is available, as is often the case in real-world deployments.
> >
> > Second, regarding the core claim that successor features (SFs) of irreversible state-action pairs are substantially lower than those of reversible pairs, I appreciate the additional empirical visualizations provided. However, given the centrality of this property to the design and functioning of RSA, a more systematic and ideally quantitative analysis would be highly beneficial. While empirical evidence and qualitative descriptions are helpful, a theoretical discussion or, at minimum, a statistical comparison (e.g., distributions of SF values for reversible vs. irreversible pairs in representative environments) would both substantiate the claim and offer practical guidance for tuning the reversibility threshold in new tasks. Such an analysis would not only enhance the rigor but also make the proposed method more accessible and robust for practitioners seeking to adapt RSA to novel domains.

---

> > > ### Author Response · Authors · 2025-08-04
> > >
> > > **[Q7] Quantitative analysis of successor features (SFs):** Thank you for your constructive feedback. We fully agree with your comment that quantitative analysis can both substantiate our claim and offer practical guidance in irreversible tasks. **Following your suggestion, we conducted additional experiments comparing the distributions of successor features (SFs) between reversible and irreversible state-action pairs in two navigation tasks and one manipulation task, all of which include irreversible states.** The distributions are represented as histogram-style bar charts, where the x-axis indicates the range of SFs and the y-axis shows the number of pairs whose SFs fall within each bin. As you pointed out, we believe that comparing the distributions of SFs between reversible and irreversible state-action pairs is an effective quantitative approach for highlighting the key contribution of our paper. We observed two expected results and two unexpected but interesting outcomes.
> > >
> > > The first expected result is that, in the early stages of training, the distributions of SFs for reversible and irreversible state-action pairs largely overlap. As training progresses, the distributions shift in opposite directions: the distribution for irreversible pairs moves toward negative values, and that for reversible pairs moves toward positive values, leading to a clear separation. Although there is some overlap between the two distributions, we provide a more detailed analysis in a subsequent paragraph (the first unexpected result).
> > >
> > > The second expected result is that, across all tasks, reversible and irreversible state-action pairs can be effectively distinguished using a reversibility threshold within a common range (between –2 and 0). This is because the feature vectors used in SFs are normalized using a running mean and standard deviation. Such normalization is task-agnostic and commonly used in reinforcement learning algorithms to train unified networks across different tasks.
> > >
> > > The first unexpected result is that, while the distribution of SFs for irreversible pairs remains stably below the reversibility threshold, a portion of the distribution for reversible pairs also falls below it. This could be interpreted as some reversible pairs being incorrectly identified as irreversible. Interestingly, our qualitative analysis revealed that these misidentified reversible states tend to be located very close to truly irreversible states, from which the agent can easily transition into an irreversible state. This conservative identification is actually beneficial to the agent, as it enables the agent to proactively avoid irreversible states, which is one of the primary objectives of RSA.
> > >
> > > The second unexpected result is that states the agent never or rarely visited, typically due to nearby irreversible states, are also identified as irreversible. Identifying such states as irreversible is intuitively reasonable, as the agent has insufficient experience in these states. This result aligns with the qualitative outcomes reported in PAINT [2] (Figure 5) for maze navigation tasks. However, we acknowledge that this identification may not be generalizable, as it blindly identifies unknown states as irreversible. To address this limitation, we will explore the use of foundation models, such as large language models (LLMs) or vision-language models (VLMs), to identify irreversible states in future work. We expect that the common-sense reasoning abilities of foundation models will make them promising for identifying irreversible states across diverse tasks, even in the early stages of training.
> > >
> > > We will add the additional quantitative results and corresponding analysis to Appendix C in the revised paper. We believe this response addresses your concerns regarding the quantitative experiments.
> > >
> > > ---
> > >
> > > [1] Archit Sharma*, Rehaan Ahmad*, Chelsea Finn. “A State-Distribution Matching Approach to Non-Episodic Reinforcement Learning.” International Conference on Machine Learning. 2022.
> > >
> > > [2] Annie Xie*, Fahim Tajwar*, Archit Sharma*, Chelsea Finn. “When to Ask for Help: Proactive Interventions in Autonomous Reinforcement Learning.” Advances in Neural Information Processing Systems. 2022.

---

> > > > ### Comment · Reviewer_wsCV · 2025-08-05
> > > >
> > > > Thank you for accepting the suggestions and conducting the additional experiments. The new empirical comparisons with MEDAL and PAINT under clearly stated assumptions provide a much more comprehensive evaluation of RSA, and the detailed quantitative analysis of successor features (SFs) distributions between reversible and irreversible state-action pairs offers valuable insight into the core mechanism of your approach. The unexpected findings regarding the conservative identification of reversible states near irreversible regions and the treatment of rarely visited states are both interesting and relevant, and your discussion of possible future directions demonstrates a thoughtful and forward-looking research agenda. These substantial additions and clarifications have alleviated my previous concerns and strengthened the paper. I am therefore raising my score.

---

> ### Author Response · Authors · 2025-08-04
>
> We again deeply appreciate your thoughtful comments, which have been very helpful in improving our work. We are pleased that many of your key questions have been clarified, and we would like to address your remaining concerns and suggestions below.
>
> ---
>
> **[Q6] Comparing MEDAL and PAINT:** We definitely agree with your comment that comparing MEDAL [1] and PAINT [2] under clearly stated assumptions would not only highlight RSA’s unique strengths but also clarify the practical trade-off. **Following your suggestion, we conducted additional experiments to compare RSA with MEDAL and PAINT on diverse maze navigation and manipulation tasks.** As with LNT-MG and R3L-MG, we assume that MEDAL and PAINT sample goals randomly and periodically from their buffers. MEDAL requires demonstrations from the optimal policy to ensure diverse initial states. While MEDAL cannot detect irreversible states, PAINT can identify and avoid them with access to reversibility labels.
>
> To collect demonstrations, we obtained the optimal policy by using SAC with repetitive resets, and executed it with diverse initial and goal states used in our evaluation. Note that SAC is the state-of-the-art RL algorithm used to implement both RSA and other baselines. To generate reversibility labels, we analyzed irreversible states for each task and defined them as follows:
>
> - PointIMaze-Trap
>
>     PointIMaze-Trap is a variant of PointIMaze that includes irreversible states. The goal is to navigate the agent to target locations as quickly as possible while avoiding purple boxes. While the agent can continue moving after colliding with brown boxes, it cannot move further after colliding with the purple boxes unless external intervention is provided. The irreversible states are then defined as those in which the agent’s x and y coordinates fall within the purple boxes.
>
> - AntOpen-Trap
>
>     AntOpen-Trap is a variant of AntOpen that introduces irreversible states. The task shares the same goal and definition of irreversible states as PointIMaze-Trap, but the agent has more complex dynamics.
>
> - HandManipulate
>
>     HandManipulate is an irreversible manipulation task. The goal is to move an egg-shaped object to target locations as quickly as possible. Dropping the object results in an irreversible state, where the agent can no longer pick up or manipulate the object without external intervention. The irreversible states are then defined as those in which the object’s z-coordinate is less than 0.05.
>
>
> We observed that MEDAL obtains higher success rates and better sample efficiency than LNT-MG and R3L-MG in reversible navigation and manipulation tasks. These results align well with the MEDAL’s underlying hypothesis that the expert demonstrations provide a good initial state distribution for learning a task. Note that PAINT behaves identically to MEDAL in irreversible tasks but differs in irreversible ones. In irreversible navigation and manipulation tasks, PAINT achieves success rates comparable to RSA while requiring fewer manual resets, suggesting that PAINT can be a preferable choice for reducing manual resets when reversibility labels are available.However, RSA obtains better sample efficiency than PAINT in both AntOpen-Trap and HandManipulate. We hypothesize that this advantage arises from the fact that, while PAINT samples goal states randomly from its buffer, RSA continuously identifies informative goal states based on the agent’s learning progress.
>
> We acknowledge that due to the short rebuttal period, the current results for MEDAL and PAINT still leave some room for improvement, particularly in terms of sample efficiency. Therefore, we will continue to conduct experiments and add the improved results to the revised paper. We hope this response addresses your concerns about the comprehensive evaluation.

---

### Official Review · Reviewer_RMHP · 2025-07-02

**Clarity:** 3
**Significance:** 2
**Originality:** 3
**Rating:** 4
**Confidence:** 3

**Summary:**

The paper proposes RSA, an ARL framework designed to reduce manual resets in both reversible and irreversible environments without relying on task-specific knowledge. RSA jointly trains a forward policy to solve tasks,  and a reset policy (to recover from failures. To avoid irreversible states, RSA encodes agent behaviors using successor features rather than relying on state-specific labels, identifying states where agents lose control over goal-relevant features. Through experiments on diverse navigation and manipulation tasks, RSA demonstrates better performance and reset efficiency compared to prior ARL methods, showing its robustness and scalability for real-world applications.

**Questions:**

1. How sensitive is RSA to the choice of feature representation $\phi$ used in successor features? Could poor feature choices lead to inaccurate identification of irreversible behaviors?

2. In large or visually rich environments, does using Random Network Distillation (RND) provide sufficient coverage for the reset policy? Would structured exploration (e.g., skill discovery) help?

3. Since RSA relies on accumulating behavioral evidence before identifying irreversible states, how can it prevent the agent from entering catastrophic states in the early stage of training?

4. Can RSA handle high-dimensional observations like images or point clouds? How would its components (e.g., state estimator, SFs) scale to such settings?

5. How accurate is the self-supervised estimator $I(s,g)$ during early training? Does misestimation bias curriculum generation?

**Ethical Concerns:**

["NO or VERY MINOR ethics concerns only"]

**Final Justification:**

The paper proposes RSA, an ARL framework designed to reduce manual resets in both reversible and irreversible environments without relying on task-specific knowledge. After the detailed rebuttal from the authors, the reviewer is convinced by the supporting answers to the proposed questions. Hence, the reviewer would maintain the preliminary rating.

**Limitations:**

yes

**Paper Formatting Concerns:**

N/A.

**Quality:**

3

**Strengths And Weaknesses:**

Strength

1. RSA avoids reliance on handcrafted rewards, reversibility labels, or demonstrations, making it more generalizable to real-world settings.

2. Using successor features to detect irreversible states via behavioral patterns is a novel approach.

3. RSA demonstrated strong performance and fewer manual resets across both reversible and irreversible tasks in a variety of domains.

Weakness

1. The reset policy uses RND, which may be inefficient in complex or high-dimensional environments.

2.  RSA can only identify irreversible states after sufficient behavioral data is collected for identification, potentially causing early-stage failures and costs.

---

> ### Author Rebuttal · Authors · 2025-07-31
>
> Thank you for your insightful and constructive feedback. We are pleased that you found the algorithm to be more generalizable to real-world settings, using of successor features to detect irreversible states to be novel, and the performance to be strong with fewer manual resets across both reversible and irreversible tasks in a variety of domains. We would like to address your comments and questions below:
>
> ---
>
> **[W1 & Q2] Using RND as the reset policy:** We definitely agree with your comment that using RND as the reset policy may be inefficient in complex environments. Your idea to leverage structured exploration with skill discovery algorithms for sufficient coverage is quite insightful.
>
> **Inspired by your idea, we found an interesting recent work called LSR [1] that utilizes skills to discover diverse initial states.** The key insight behind LSR is that the need to reset the agent with diverse initial states provides a natural setting for discovering distinct skills. We believe LSR can be a strong starting point for integrating your idea into our work. We would be glad to further explore this research direction in future work.
>
> ---
>
> **[W2 & Q3] Preventing the agent from entering irreversible states in the early stage of training:** We are glad to discuss the issue of entering irreversible states in the early stage of training. While this issue is not considered in this paper, we are developing a key idea to address it as follows: unlike informative states that should be identified based on the agent’s learning progress, irreversible states can be identified regardless of the agent’s learning progress. For example, once an agent in HandManipulate drops an object, it can no longer pick up or manipulate the object, regardless of how much it has learned.
>
> Based on our key idea, we are exploring the use of foundation models, such as large language models (LLMs) or vision-language models (VLMs), to identify irreversible states. These models are typically pre-trained on massive offline datasets and then frozen during downstream use. Therefore, they are not well-suited to estimating an agent’s learning progress to identify informative states every iteration. However, their common-sense reasoning abilities make them promising for identifying irreversible states across diverse tasks even in the early stage of training. Suppose we have access to GPT-4o, a multimodal foundation model that can handle both text and images. If we provide GPT-4o with the current image observation and a text query such as “Does this image represent a reversible state?”, it could respond with a binary answer like “yes” or “no” without requiring further training. This binary signal can then be used to identify irreversible states even in the early stage of training. We will add this discussion to Section 6 in the revised paper. We hope this discussion addresses your concern about early-stage failures.
>
> ---
>
> **[Q1] Choice of feature representation used in successor features:** Designing mapping functions that extract relevant feature representations from states or observations is critical for identifying irreversible behaviors. **We did not consider this issue as it is orthogonal to our main contribution of reducing manual resets in both reversible and irreversible environments without task-specific knowledge.** To be specific, following previous goal-conditioned RL algorithms, RSA extracts relevant feature representations by using the mapping functions provided by the tasks in our experiments.
>
> Before concluding this discussion, we would like to emphasize that combining RSA with state-of-the-art representation learning algorithms to learn the mapping functions is an interesting direction for future work. Note that RSA is agnostic to the form of the mapping function. We will modify the first paragraph of Section 4.2 to make the importance of feature representation clearer in the revised paper.
>
> ---
>
> **[Q4] Handling high-dimensional observations: RSA is fully compatible with all types of observations, including high-dimensional observations such as images and point clouds.** To handle such observations, the models within RSA should be constructed using more sophisticated networks, such as Transformers or MLP-Mixers. While training these advanced networks may require additional techniques, we expect that such techniques can be easily integrated into our work. We will add this discussion to Section 6 in the revised paper.
>
> ---
>
> **[Q5] How accurate the state information estimator is during early training:** The initialized state information estimator cannot accurately estimate how informative initial and goal states are in the early stage of training. Note that RSA does not pretrain the state information estimator, as doing so would require task-specific knowledge, such as expert demonstrations. Instead, RSA initializes the state information estimator to identify all pairs of initial and goal states as informative. We empirically observed that this simple and practical initialization trick enables the agent to collect diverse pairs of initial and goal states during early training, which alleviates biased curriculum generation. The implementation of this initialization trick corresponds to the weight_init function in utils.py of our code. We hope this discussion addresses your concern about the state information estimator.
>
> ---
>
> Thank you again for your time and effort in reviewing our work. We have carefully considered your valuable comments and tried our best to address your concerns with the clarifications and the additional new experiments. If our response has addressed your primary concerns, we would greatly appreciate it if you could update your score to reflect that. We are also willing to discuss with you if you have any further concerns.
>
> [1] Kelvin Xu, Siddharth Verma, Chelsea Finn, Sergey Levine. “Continual Learning of Control Primitives: Skill Discovery via Reset-Games.” Advances in Neural Information Processing Systems. 2020.

---

> > ### Comment · Reviewer_RMHP · 2025-08-04
> >
> > Thanks for the detailed response and discussion. I will keep my score and recommend the acceptance.

---

### Official Review · Reviewer_wpan · 2025-07-07

**Clarity:** 2
**Significance:** 2
**Originality:** 3
**Rating:** 4
**Confidence:** 4

**Summary:**

This paper proposes an autonomous RL algorithm called RSA based on two components: 1) a curriculum created by setting goal states for forward and backward policies based on which states would be informative goal/initial states. This is done with a combination of heuristics and by training a classifier that estimates the ability of the current agent to reach goals. 2) A mechanism to detect and avoid irreversible states using successor features. They show that this algorithm needs fewer resets and learns faster on various navigation and manipulation environments compared to previous methods.

**Questions:**

- What does RSA stand for? Robust and Scalable Autonomous RL? If so, can you further explain the scalable part of the name?
- Why is using the final state of failed iterations as the goal state for the next iteration a good heuristic? In HER it makes sense since the agent is able to learn without having to rollout more trajectories, but in this case, it’s not clear why those failed intermediate states would be good future goal states. Can you run an ablation to see whether this is necessary? And if so provide a clearer understanding of why?
- Is there any weighting of more recent trajectories for training the state information estimator given that sampling old trajectories could lead to out of date labels?
- I am not exactly clear on how the mapping function $m$ used in computing the successor features comes from. Section 3.1 implies that it comes from the environment and is not learned?
- Could you create a visualization of which states the agent believes are irreversible and how that evolves over time across a couple environments? It would be interesting to see how well that aligns with our notion of irreversible.
- Is get_informative_goal in general just sampling from the goal states provided by the environment (except in the cases of failed forward trajectories)?
- Can you provide the ablation results for the rest of the environments as well?
- For Figure 7, right, it would be helpful to also see the success rate plotted against the timesteps in the environment as well.

**Ethical Concerns:**

["NO or VERY MINOR ethics concerns only"]

**Final Justification:**

The authors have answered my most pressing questions. I will be maintaining my score.

**Limitations:**

Yes

**Quality:**

3

**Strengths And Weaknesses:**

Strengths:
- The algorithm presented is effective at learning in an autonomous fashion in both reversible and irreversible environments.
- The mechanism to find irreversible states seems like it works well.
- The idea of switching from the forward to the reset policy once the initial state is difficult enough is interesting.

Weaknesses:
- The method does introduce 3 new hyperparameters that need to be searched over. It’s also unclear how the search was done, and if the baselines were searched over in a commensurate fashion.
- The goal setting heuristic (when the policy fails at achieving the forward goal) seems not as well motivated. It’s unclear why using the last state of the failed rollout is a good forward goal. A useful ablation to have would be if you simply chose a goal from the rest of your goal set.
- Some parts of the detecting irreversible states method are unclear (see below).

---

> ### Author Rebuttal · Authors · 2025-07-31
>
> We would like to thank you for taking the time to review our work and for providing thoughtful suggestions. We are pleased that you found the algorithm to be effective in both reversible and irreversible environments, the mechanism for finding irreversible states to work well, and the idea of switching between the forward and reset policies to be interesting. Below are the responses regarding the raised concerns and suggestions. We believe that our responses address all issues in the review.
>
> ---
>
> **[W1] How the hyperparameters are searched:** Thank you for your constructive feedback. To provide a fair comparison, **we performed grid search for the hyperparameters in both RSA and the baselines, and tried our best to find the optimal values.** Table 2 in Appendix B summarizes the key hyperparameters used in our experiments.
>
> To address your concern, **we also conducted additional experiments on PointIMaze-Trap to examine the effects of our key hyperparameters.** First, we analyzed the performance according to the hyperparameters, $\lambda_1$ and $\lambda_2$, which are the lower and upper thresholds of the reachability range of informative initial or goal states. We observed that too difficult initial and goal states ($\lambda_1=0.0$ and $\lambda_2=0.1$) cause even more manual resets than random initial states. Conversely, too easy initial and goal states ($\lambda_1=0.9$ and $\lambda_2=1.0$) result in lower sample efficiency than the informative initial and goal states ($\lambda_1=0.1$ and $\lambda_2=0.6$). These results confirm the benefits of identifying informative initial and goal states.
>
> Next, we investigated the performance according to the reversibility threshold, $\lambda_3$. We observed that RSA maintains good performance and properly identifies irreversible states, except when the reversibility threshold is much higher than the output of the initialized successor features, $\psi^{\pi_r}(s,a)$, which causes the agent to classify all states as irreversible at the beginning of training. This discussion and additional experiments will be added to Appendix C in the revised paper.
>
> ---
>
> **[W2 & Q2] Using the last state of the failed rollout as the goal state for the next iteration:** The last state of the failed rollout and its neighboring states are clearly not trained enough for at least one goal state, including the current goal state that the agent failed to reach. Placing a goal state near these states can provide meaningful learning signals to them. We empirically found that using the last state of the failed rollout as the goal state for the next episode is a simple and efficient way to provide such signals without any additional computation.
>
> Following your constructive suggestion, **we conducted additional experiments to examine the benefits of using the last state of the failed rollout as the next goal state.** We compared RSA to the following two variants: 1) randomly sampling the next goal state from all previous goal states, and 2) randomly sampling the next goal state from the identified set of informative goal states. As expected, RSA outperforms the first variant and achieves comparable performance with the second variant while requiring less computational cost. We will update the fourth paragraph of Section 4.1 to provide a clearer understanding of using the last state of the failed rollout as the next goal state, and add additional experimental results to Appendix C in the revised paper.
>
> ---
>
> **[W3 & Q4] The mapping function $m$ used in computing the successor features:** Following previous goal-conditioned RL algorithms, we use mapping functions provided by the tasks in our experiments. While learning the mapping functions is orthogonal and complementary to the main contribution of our work, we believe that combining RSA with state-of-the-art representation learning algorithms to learn these functions is an interesting direction for future work. Note that RSA is agnostic to the form of the mapping function. We will add this discussion to Section 6 in the revised paper.
>
> ---
>
> **[Q1] What RSA stands for:** RSA refers to the Robust and Scalable Autonomous (RSA) RL algorithm introduced in this paper. We originally denoted the term as RSA-RL but decided to use the more concise form, RSA, for brevity and readability.
>
> RSA is more scalable than previous ARL algorithms, which typically assume either reversible environments or access to task-specific knowledge, such as demonstrations or reversibility labels, for identifying irreversible states. These assumptions are impractical and difficult to scale to real-world tasks, where each task may have its own unique set of irreversible states. **To the best of our knowledge, RSA is the first autonomous RL algorithm that can reduce manual resets in both reversible and irreversible environments without task-specific knowledge.** Table 1 in the main paper summarizes the key features that distinguish RSA from previous ARL algorithms.
>
> ---
>
> **[Q3] Weighting of more recent trajectories for training the state information estimator:** We did not use the idea of weighting more recent trajectories in our experiments. Specifically, Figures 5 and 6 in the main paper, which demonstrate that the state information estimator can identify informative initial and goal states based on the agent’s learning progress, were obtained by sampling trajectories randomly.
>
> As a side note, we find the idea of weighting more recent trajectories to be quite interesting. We believe this idea could make the training of the state information estimator more efficient and stable. Therefore, we would be glad to further explore this idea in our future work.
>
> ---
>
> **[Q5] Visualizing how irreversible states evolve over training time:** Thank you for your constructive and insightful feedback. We definitely agree with your comment that visualizing how irreversible states evolve over training time would be interesting. Following your suggestion, **we implemented additional evaluation code to visualize irreversible states in a manner similar to Figures 5 and 6.** The visualization results on PointIMaze-Trap and AntOpen-Trap show that at the beginning of training, all states are identified as reversible, as the state information estimator is initialized to output values higher than the reversibility threshold. As training progresses, irreversible states are gradually detected around the edge of the purple boxes, which represent the boundaries between reversible and irreversible regions of the state space. These results indicate that irreversible states identified by RSA align well with our intuitive notion of irreversibility. We will add the visualization results to Appendix C in the revised paper.
>
> ---
>
> **[Q6] get_informative_goal function:** We are pleased to discuss the get_informative_goal$(B_r, G_{1:k-1}, I(s,g))$ function, presented in Algorithm 1 of the main paper. This function outlines how RSA takes into account the agent’s learning progress to select an informative goal state $g_k$ at iteration $k$. **We would like to emphasize that selecting informative goal states is fundamentally distinct from randomly sampling ordinary goal states from the predefined set provided by the environment.** RSA utilizes the state information estimator to identify a subset of informative goal states from all goal states encountered in previous forward rollouts, including both successful and failed rollouts. It then samples one informative goal state for the current episode from this identified subset. This process prevents the agent from being assigned goal states that are either too difficult or too easy. Note that, unlike previous ARL algorithms, RSA does not assume that specific candidate goal states are predefined by the environment but rather allows goal states to be located anywhere within the entire state space. **Algorithm 2 in Appendix A describes the procedure of get_informative_goal$(B_r, G_{1:k-1}, I(s,g))$ in further detail.**
>
> ---
>
> **[Q7] Ablation results for the rest of the environments:** To address your concern, **we conducted additional ablation studies to examine the benefits of the key components of RSA in maze navigation tasks.** First, we compared RSA with its variant, RSA w/o SIE (which does not identify informative states), in the reversible environments PointIMaze and AntOpen. Similar to the left plot of Figure 7, RSA achieves better asymptotic performance and sample efficiency than RSA w/o SIE. This result confirms the benefits of identifying informative initial and goal states with SIE.
>
> Next, we compared RSA with its variant, RSA w/o SFs (which does not identify irreversible states), in the irreversible environments PointIMaze-Trap and AntOpen-Trap. Unsurprisingly, RSA w/o SFs failed to learn both navigation tasks as it is unable to detect and avoid irreversible states. This outcome is consistent with the results shown in the right plot of Figure 7. We will update Figure 7 to include these additional ablation results and add the corresponding discussion to the fourth paragraph of Section 5.2 in the revised paper.
>
> ---
>
> **[Q8] Plotting the success rate against timesteps (Figure 7):** We definitely agree with your comment that including a plot of the success rate against timesteps, even in the irreversible environments, would be helpful in improving our work. Changing the x-axis of the plot poses no difficulty and does not require additional effort. We will add the plot against timesteps to Appendix C in the revised paper.
>
> ---
>
> Thank you again for your time and effort in reviewing our work. We have carefully considered your valuable comments and tried our best to address your concerns with the clarifications and the additional new experiments. If our response has addressed your primary concerns, we would greatly appreciate it if you could update your score to reflect that. We are also willing to discuss with you if you have any further concerns.

---

> > ### Comment · Reviewer_wpan · 2025-08-06
> >
> > The authors have answered my most pressing questions. I will be maintaining my score. I think the main thing I'd like to see improved for the final version other than the visualizations mentioned is a bit more clarity on the specifics of the algorithm and experiments, such as how you seed the initial goal states (G_0) and the details of the grid used for the hyperparameter search, just to make it easy for future reproduction.

---

> ### Author Response · Authors · 2025-08-07
>
> We are glad to have another opportunity to engage in further discussion with you. Your constructive comments have been significantly helpful in improving our work. We are relieved that our previous response addressed your pressing questions. However, as you pointed out, we acknowledge that some details of the algorithm and experiments may have been insufficient due to the page limit. **To provide greater clarity, we offer detailed explanations in the following paragraphs for the two ambiguous points you raised, along with three additional clarifications.** We hope our response fully addresses your remaining concerns.
>
> ---
>
> **[Q9] Details of grid search:** Thank you for your thoughtful feedback. We definitely agree with your comment that providing more clarity on the details of the grid search will facilitate future reproduction. We use different strategies to determine the search ranges for standard RL hyperparameters and those newly introduced in RSA. Specifically, RSA introduces three hyperparameters: $\lambda_1$ and $\lambda_2$, which define the lower and upper thresholds of the reachability probability over informative states, and $\lambda_3$, which serves as the reversibility threshold for identifying irreversible states.
>
> For standard RL hyperparameters, we use search ranges commonly found in existing RL algorithms. In particular, the learning rate is searched over {0.005, 0.003, 0.001, 0.0005, 0.0003, 0.0001}, and the batch size over {64, 128, 256, 512}. Note that the forward and reset agents can be trained with different learning rates and batch sizes.
>
> For the hyperparameters introduced in RSA, we search $\lambda_1$ over {0.1, 0.2, 0.3, 0.4}, $\lambda_2$ over {0.6, 0.7, 0.8, 0.9}, and $\lambda_3$ over {–3.0, –2.5, –2.0, –1.5, –1.0, –0.5, 0.0}. These search ranges are clearly more efficient and distinct from searching over the entire valid range, such as from 0.0 to 1.0 or from –3.0 to 3.0. **It is the following key hypotheses of RSA that allow us to define the efficient search ranges.**
>
> First, the hypothesis underlying the search ranges for $\lambda_1$ and $\lambda_2$ is that informative states are those that are neither overly difficult nor trivially easy for the agent. In other words, RSA identifies informative states as those with neither too low nor too high reachability probability. This suggests that the entire range from 0.0 to 1.0 need not be searched to tune these hyperparameters. The additional experiments presented in our response to [W1] show that too difficult initial and goal states ($\lambda_1=0.9$ and $\lambda_2=1.0$) cause more manual resets than randomly sampled states, and too easy initial and goal states ($\lambda_1=0.0$ and $\lambda_2=0.1$) lead to lower sample efficiency than informative states. Note that $\lambda_1$ and $\lambda_2$ are set to the same values across all tasks in our experiments.
>
> Second, the hypothesis behind the search range for $\lambda_3$ is that the successor features (SFs) for irreversible state-action pairs are much lower than those for reversible pairs. This claim is based on our empirical observation that, in irreversible states, an agent loses control over goal-relevant features, regardless of the actions it takes. To validate our claim, we conducted additional experiments comparing the distributions of SFs between reversible and irreversible state-action pairs. The experimental results confirm that SFs for irreversible pairs are lower than those for reversible pairs. We also observed that, across all tasks, the irreversible states can be effectively distinguished using a reversibility threshold within a common range (between –2 and 0). This is because the feature vectors used to calculate SFs are normalized using a running mean and standard deviation. Such normalization is task-agnostic and commonly used in RL algorithms to train unified networks across different tasks. Further details of these experiments are provided in our response to [Q7] from reviewer “wsCV”.
>
> Before concluding this discussion, **we would like to emphasize that optimizing the RSA hyperparameters does not require any task-specific knowledge. This advantage allows us to use the same hyperparameter values across diverse tasks, significantly reducing the burden of tuning.** We will add this discussion to Appendix B in the revised paper. We believe this discussion addresses your concerns about the search grid.

---

> ### Author Response · Authors · 2025-08-07
>
> **[Q10] How to seed initial goal states:** Thank you for pointing out what we missed in our paper. Since no trajectories have been stored at the time of sampling the initial goal states ($G_0$), we rely on a random state sampling function provided by the tasks to choose the initial goal states. This function either samples uniformly from the entire state space or from a predefined set of candidate goal states. To ensure a fair comparison, the initial goal states for both RSA and all baselines are sampled in the same way using this function. We will add this discussion to Section 4.1 in the revised paper. We hope this response addresses your concern about initial goal states.
>
> As a side note, RSA utilizes a warmup period during which the agent takes random actions to explore environments, and no updates are performed. Leveraging such a warmup period is a common technique in RL to stabilize early training. During this period, RSA does not rely on the state information estimator, as it has not yet been trained, and instead samples goal states randomly from all goal states in the buffer. This design choice enables the agent to collect diverse and unbiased trajectories. For more details, please refer to the run_train function in train.py of the submitted code. We will add this discussion to Appendix A in the revised paper.
>
> ---
>
> **[Additional Clarification 1] Details of get_informative_goal function:** The get_informative_goal function, presented in Algorithm 1 of the main paper, is critical to understanding how RSA works. Its details are currently provided in Appendix A due to the page limit. **Since the camera-ready version allows for an additional content page, we will add a new section (Section 4.3 in the revised paper) that includes the pseudo code and a corresponding description of this function.** We believe this update will help deepen readers’ understanding of how RSA works.
>
> Note that the pseudo code for get_informative_goal$(B_r, G_{1:k-1}, I(s,g))$ function is currently shown in Algorithm 2 of Appendix A. The corresponding description is as follows: RSA first checks whether the previous goal state $g_{k−1}$ was achieved. If so, it samples a batch of $N_s$ initial states from the reset buffer $B_r$ and computes the average probability of reaching $g_{k−1}$ from those states. If the average probability lies within the range [$\lambda_1$, $\lambda_2$], the goal state $g_{k−1}$ is considered still informative and is reused. Otherwise, RSA samples $N_g$ candidate goal states from the previously collected goal state set $G_{1:k-1}$ to search for a new informative goal state. For each candidate goal $\bar{g}$, RSA computes the average probability of reaching it from the sampled initial states and returns the first goal state whose probability lies within the same range. If the previous goal state $g_{k−1}$ was not achieved, RSA simply reuses $g_{k−1}$ as the goal state for the kth iteration.
>
> ---
>
> **[Additional Clarification 2] How to determine initial states before training state information estimator:** We found that the procedure for determining initial states before training the state information estimator could be further clarified. As discussed in our response to [Q9], RSA begins with a warmup period during which the agent takes random actions, and no updates are performed. At this period, the state information estimator is not sufficiently trained to estimate how informative initial or goal states are. Using it prematurely could lead to biased identification and sub-optimal performance. **To avoid this issue, during the warmup period, RSA activates the reset policy until the maximum episode step and uses the final state as the initial state, without using the state information estimator to identify informative ones.** This design choice enables the agent to collect diverse and unbiased experiences. We observed that this is enough to achieve significant performance gains over previous state-of-the-art ARL algorithms across various navigation and manipulation tasks. We will add this discussion to Appendix A in the revised paper.
>
> We also acknowledge that activating the reset policy until the maximum episode step without identifying informative states can increase the risk of entering irreversible states during the warmup period. To address this limitation, we will explore the use of foundation models, such as large language models (LLMs) or vision-language models (VLMs), to identify irreversible states in future work. We expect that the common-sense reasoning abilities of foundation models will make them promising for identifying irreversible states across diverse tasks, even in the early stages of training. We will add this discussion to Section 6 in the revised paper.

---

> ### Author Response · Authors · 2025-08-07
>
> **[Additional Clarification 3] Initialization of state information estimator and successor features:** We would like to clarify how the state information estimator and the successor features are initialized. In the early stages of training, both models may struggle to accurately identify informative or irreversible states, which can lead to biased exploration. **To mitigate this issue, we initialize both models to identify all states as informative and reversible.** We empirically observed that this simple initialization technique is highly effective for collecting diverse and unbiased trajectories and stabilizes early training. We will add this discussion to Appendix B in the revised paper.
>
> ---
>
> We carefully considered your valuable comments and did our best to provide clearer descriptions of our algorithm and experiments. We believe our response addresses your concerns, and would be happy to discuss further if you have any additional questions or concerns.

---

### Note · Authors · 2025-08-13

We would like to thank all reviewers for taking the time to review our work and for providing thoughtful comments. We are pleased that the reviewers found the motivation to be crystal clear, the problem addressed to be significant, the algorithm to be novel and well-grounded, the performance to be strong with fewer manual resets across both reversible and irreversible tasks in various domains, and the submitted code to be well-documented.

**To the best of our knowledge, the proposed algorithm, called RSA, is the first autonomous reinforcement learning (ARL) algorithm that can reduce manual resets without relying on task-specific knowledge in both reversible and irreversible environments.** It stands in stark contrast to previous ARL algorithms that rely on task-specific knowledge, such as expert demonstrations or reversibility labels, to reduce manual resets. **Experimental results demonstrate that, in diverse maze navigation and manipulation tasks, RSA achieves more robust performance and better sample efficiency than previous state-of-the-art ARL algorithms, while requiring fewer manual resets.** Given that the primary bottleneck in learning real-world tasks is manual resets, which completely halt data collection, we believe RSA represents an important step toward scaling RL algorithms to real-world tasks.

Following the insightful suggestions of the reviewers, we conducted additional quantitative and qualitative experiments, including ablation studies. The experimental results confirm the benefits of the key components of RSA and demonstrate that RSA can be extended to stochastic environments. As the reviewers noted, these results substantiate the key ideas underlying RSA: 1) generating a curriculum by identifying informative initial and goal states based on the agent’s learning progress, and 2) detecting and avoiding irreversible states by encoding the behaviors exhibited in those states rather than the states themselves. These key ideas are fundamentally different from those of previous algorithms. In addition to the experiments, we also provided clearer descriptions of our algorithm and experiments, including a discussion of theoretical insights and training complexity.

We will make sure that the revised paper includes the clarifications and additional experiments described in our responses. We believe that our responses address all issues and concerns of the reviewers, and we would appreciate it if you could take the final remarks into account.

---

### Decision · Program_Chairs · 2025-09-17

**Decision:**

Accept (poster)

**Comment:**

A. This paper proposes an RL method to minimize manual resets by learning a forward and backward policy. The method uses successor features to identify states where agents lose control over goal-relevant features. During training, the method uses a curriculum for learning. Experiments show that the proposed method requires fewer resets and learns faster on various navigation and manipulation environments compared to previous methods.

B. Reviewers appreciated that the method lifts assumptions of prior work (e.g., no hand-crafted rewards) while achieving strong empirical performance. They also appreciated that the paper included code.

C. Reviewers had questions about a few aspects of the paper (e.g., computational costs, performance in stochastic environments) and suggested some new experiments and baselines (e.g., MEDAL, PAINT); these concerns were addressed through the discussion and additional experiments.

D. Reviewers unanimously voted to accept the paper.

E. Reviewers conducted new experiments (e.g., stochastic settings, MEDAL and PAINT baselines). Reviewers confirmed that authors answered questions.